# Protective Efficacy of a Mucosal Influenza Vaccine Formulation Based on the Recombinant Nucleoprotein Co-Administered with a TLR2/6 Agonist BPPcysMPEG

**DOI:** 10.3390/pharmaceutics15030912

**Published:** 2023-03-10

**Authors:** Maria Victoria Sanchez, Thomas Ebensen, Kai Schulze, Diego Esteban Cargnelutti, Eduardo A. Scodeller, Carlos A. Guzmán

**Affiliations:** 1Laboratorio de Inmunología y Desarrollo de Vacunas, Instituto de Medicina y Biología Experimental de Cuyo (IMBECU), CCT-CONICET, Universidad Nacional de Cuyo, Mendoza M5500, Argentina; vicky_sanchez_@hotmail.com (M.V.S.); diegocargnelutti@hotmail.com (D.E.C.); eduardo_scodeller@hotmail.com (E.A.S.); 2Department of Vaccinology and Applied Microbiology, Helmholtz Centre for Infection Research, 38124 Braunschweig, Germany; thomas.ebensen@helmholtz-hzi.de (T.E.); kai.schulze@helmholtz-hzi.de (K.S.)

**Keywords:** BPPcysMPEG, MALP-2, influenza, mucosa, adjuvant, vaccine, nucleoprotein, TLR2/6 agonist

## Abstract

Current influenza vaccines target highly variable surface glycoproteins; thus, mismatches between vaccine strains and circulating strains often diminish vaccine protection. For this reason, there is still a critical need to develop effective influenza vaccines able to protect also against the drift and shift of different variants of influenza viruses. It has been demonstrated that influenza nucleoprotein (NP) is a strong candidate for a universal vaccine, which contributes to providing cross-protection in animal models. In this study, we developed an adjuvanted mucosal vaccine using the recombinant NP (rNP) and the TLR2/6 agonist S-[2,3-bispalmitoyiloxy-(2R)-propyl]-R-cysteinyl-amido-monomethoxyl-poly-ethylene-glycol (BPPcysMPEG). The vaccine efficacy was compared with that observed following parenteral vaccination of mice with the same formulation. Mice vaccinated with 2 doses of rNP alone or co-administered with BPPcysMPEG by the intranasal (i.n.) route showed enhanced antigen-specific humoral and cellular responses. Moreover, NP-specific humoral immune responses, characterized by significant NP-specific IgG and IgG subclass titers in sera and NP-specific IgA titers in mucosal territories, were remarkably increased in mice vaccinated with the adjuvanted formulation as compared with those of the non-adjuvanted vaccination group. The addition of BPPcysMPEG also improved NP-specific cellular responses in vaccinated mice, characterized by robust lymphoproliferation and mixed Th1/Th2/Th17 immune profiles. Finally, it is notable that the immune responses elicited by the novel formulation administered by the i.n. route were able to confer protection against the influenza H1N1 A/Puerto Rico/8/1934 virus.

## 1. Introduction

The vast majority of the current vaccines used to fight seasonal influenza outbreaks are formulated with antigens from inactivated viruses and administered by the intramuscular (i.m.) route. Thereby, vaccine efficiency is based on the induction of neutralizing antibodies directed against an antigen, such as the viral hemagglutinin (HA) [1]. This type of immunity is mainly specific for each virus strain included in the vaccine, therefore, its effectiveness decreases significantly against other heterologous strains [2]. One major handicap of parenterally administered vaccines is that they do not induce significant mucosal and T cell immune responses, which are very important in order to induce effective protection in the human population [3,4,5]. This scenario becomes more problematic in case of the emergence of new pandemic strains for which the world’s population does not have any type of immunological coverage [6]. In order to overcome these hurdles, multiple strategies have been tested to generate new influenza vaccines:(i)improvement of the performance of current vaccines with new adjuvants and alternative routes of vaccine administration [7,8,9,10,11,12,13],(ii)development of new recombinant antigens able to promote the elicitation of cross-neutralizing antibodies to generate sterilizing immunity against all influenza strains [14,15,16,17,18,19,20], and(iii)utilization of relatively conserved influenza antigens (e.g., nucleoprotein) capable of inducing strong T cell responses that display a high degree of cross-reactivity against various influenza strains, [13,21,22,23,24].

Several conserved antigens have been identified, such as the ectodomain of viral protein M2 (M2e), the M1 protein, and the viral NP, which are able to induce protection against a broader range of influenza strains mediated by non-sterilizing immunity [25,26,27].

When viral NP is used for the development of a universal T cell vaccine, protection is mediated by the elimination of infected cells through the action of NP-specific CD8^+^ cytotoxic T-lymphocytes [28,29,30]. Antigen-specific CD8^+^ T cells are known to contribute to protection against influenza virus infections by limiting the duration and severity of the disease [22]. However, there is growing evidence that CD4^+^ T-lymphocytes also play a remarkable role in the protection against infection. In several studies, it has been demonstrated that the induction of robust CD4^+^ T cell memory responses is important to provide help to B-lymphocytes, whereby conserved MHC Class II-restricted epitopes within HA are essential for B cells to respond to drifting influenza [31]. Mice studies have also revealed that even non-neutralizing NP-specific antibodies can help T-lymphocytes to induce protective immune responses. Thus, anti-NP IgG antibodies cooperate by forming immuno-complexes with the antigen, which are then recognized by the Fc receptors on cells of the innate immune system, such as dendritic cells and macrophages, enhancing the activation and presentation of antigens to T-lymphocytes [32,33,34]. LaMere et al., also highlighted the correlation between the anti-NP IgG antibody titer and the long-term hetero-subtype protection [33].

In addition, memory CD4^+^ T cells not only support B cells but also cytotoxic responses during influenza infection [35]. CD4^+^ influenza-specific T cells also support specific CD8^+^ T cells while maintaining their cytolysis activity by producing IFNγ and perforin [31,36,37,38]. Moreover, it has been demonstrated that memory CD4^+^ T cells recognizing immunodominant epitopes of the M1 and NP proteins were associated with the reduction in influenza symptoms and the limitation of viral replication in experimentally infected humans [39].

It is important to note that in the case of a possible influenza pandemic, there is still an urgent need to produce safe, cost-effective, and scalable universal vaccines that can be produced in developing or low- and middle-income countries. Recombinant proteins offer the advantage of rapid manufacture but often have poor immunogenicity. Therefore, the incorporation of adjuvants is necessary to stimulate efficient antigen-specific immune responses. Several adjuvants, such as toll-like receptor agonists such as TLR2, TLR4, TLR7/8, and TLR9, squalene oil-in-water mixtures, or stimulators of interferon genes (STING) agonists, have been exhaustively investigated for their ability to elicit efficient humoral and cellular responses; however, there is limited knowledge on the ability of mucosal adjuvants to induce effective immunity [40].

It has been emphasized in recent years that the generation of an effective immune response on mucosal surfaces is important for protection against respiratory tract infections [41,42]. For vaccines aiming to protect the body against pathogens that enter the host through the airways, the mucosal route should be preferred in comparison to parental administration [43]. Thus, the vaccination via mucosal application, e.g., intranasal (i.n.) route, is able to induce systemic but also local immune responses on mucosal surfaces, which play a major role in the combat against influenza infection. For example, there is a significant contribution of mucosal antigen-specific IgA antibodies as well as lung-resident cellular immunity to influenza virus clearance [44]. Thereby, Askovich et al., provided experimental data that early activation of Il-17 production correlates with increased protection against influenza virus challenge in mice [45,46]. Furthermore, it has been demonstrated that frequencies of Th17 and Tc17 cells mediate protective immunity against highly virulent influenza strains in humans [47].

The aim of this study was to develop a universal vaccine capable of rapid and easy production and capable of inducing efficient mucosal and cellular responses that confer protection against influenza infection. For this purpose, the vaccine was designed with the NP antigen, which was obtained as recombinant protein, and it was adjuvanted with a TLR2/6 agonist, the BPPcysMPEG adjuvant [48,49]. There are some limitations for the parenteral compound Malp-2, such as poor biosolubility in liquid solutions. Moreover, this compound is able to stimulate the immune system by strong induction of various inflammatory mediators, which can influence the biocompatibility. To overcome these limitations, various physicochemical properties and surface modification strategies have been employed for the novel BPPcysMPEG, such as pegylation. In contrast to aluminium-adjuvanted vaccines, which have certain limitations, such as no Th1 reactivity and low stability at low temperatures, BPPcysMPEG showed high stability at low temperatures, enhanced biosolubility in liquid solutions, and was able to induce both antigen-specific cellular (Th1) and humoral (Th2) immune responses. Moreover, the synthetic TLR2/6 ligand BPPcysMPEG, a pegylated synthetic derivative of the macrophage-activating lipopeptide 2kDa (MALP-2) [50,51,52], is a powerful adjuvant capable of promoting enhanced immune responses given by the mucosal route [53,54,55]. Previous studies have shown that antigens co-administered with MALP-2 by the mucosal route induced an enhanced B- and T- cell response and improved antigen presentation by dendritic cells, similar or even superior to those observed following parenteral immunization with the same formulations [51,52,56,57]. In comparison to the parent compound MALP-2, BPPcysMPEG showed improved water solubility while retaining its agonistic capacity to stimulate the TLR-2/6 heterodimer [58,59,60,61]. Furthermore, MALP-2 has been described to exert beneficial effects on organ damage and the further course of trauma and sepsis [62]. These findings lead us to believe that BPPcysMPEG could be a safe adjuvant candidate for influenza vaccines, which would be capable of inducing mucosal immunity, T cell responses, and humoral immunity.

The studies described here evaluate the immunogenicity and the protection efficacy of rNP plus BPPcysMPEG in a murine model. Moreover, our vaccine candidate not only stimulated protective immune responses in mice following application by the more classical parenteral route but also following intranasal administration, highlighting its intrinsic potential.

## 2. Material and Methods

### 2.1. Vaccine Design

The nucleoprotein gene derived from influenza strain A/PR/8/34 (H1N1) was cloned into the pET30a plasmid, and the protein was expressed in *Escherichia coli* BL21, (DE3) bacteria, purified, and LPS decontaminated. The recombinant NP was used in previous works by Cargnelutti et al. [48,49]. The BPPcysMPEG, International Patent Classification (IPC): A61K 47/48 (2006.01), Pub. No.: WO/2007/059931, a pegylated derivative of MALP-2, was synthesized at the Helmholtz Centre for Infection Research (HZI), Braunschweig, Lower Saxony, Germany. 38 μmol (34 mg) of fluorenylmethoxycarbonyl (Fmoc)-protected BPPcys (Pam2cys) compound was dissolved in 25 mL dimethylformamide (DMF, (reagent-grade, 7032) J.T. Baker, Deventer, Netherlands), #7032 di-chlormethan (DCM, 7053, J.T. Baker, Deventer, Netherlands) in a 2:1 ratio. Subsequently, 38 μmol (36 µL) di-isopropylcarbodiimide (DIC, Sigma-Aldrich Chemie GmbH, Taufkirchen, Germany), #38370 and 38 μmol (6 mg) of anhydrous hydroxylbenztriazole (HOBt, #A28536, Merck Millipore, Darmstadt, Germany) was added to the solution. Afterwards, 38 μmol (76 mg) monomethoxy-amino-PEG (Mw: 5000 Da, Rapp Polymere GmbH, Tuebingen, Germany), #12 5000-2 was added to the mixture and incubated for 24 h at RT. After the separation of the Fmoc-protection group with 10 mL (20%) piperidine (Fluka, #80640)—DMF solution for 15 min at RT, the compound was purified by column chromatography with silica gel 60 (Merck Millipore (Darmstadt, Germany, #9385)) and DCM—methanol in a ratio of 95:5 and 90:10. The resulting BPPcysMPEG compound was characterized by Maldi-MS and NMR-spectral analysis. The different synthesis steps for the production of BPPcysMPEG were described in detail in different articles [63,64]. To rule out LPS contamination during the production process, each BPPcysMPEG batch was analyzed by the HEK-Blue™ LPS Detection Kit (#rep-lps2, InvivoGen EUROPE, Toulouse, France).

### 2.2. Mice

Six- to eight-week-old female BALB/c (H-2d) mice (Harlan Winckelmann GmbH, Borchen, Germany) were bred at the animal facility of the Helmholtz Centre for Infection Research under specific pathogen-free (SPF) conditions. For vaccination studies, groups of 5 mice were immunized with 10 μg/dose rNP derived from the influenza strain A/PR/8/34 (H1N1) alone or combined with 10 μg/dose BPPcysMPEG (HZI research-grade quality), administered by the i.n. (20 μL) or s.c. (100 µL) route on days 0 and 21 [48,49]. Control mice were vaccinated with PBS. To facilitate i.n. immunization, mice were briefly anesthetized with isofluorane (Abbott Animal Health, Chicago, IL, USA).

### 2.3. Ethics Section

All animal experiments, including the sublethal challenge with influenza strain A/Puerto Rico/8/34 (H1N1), were approved by (i) the animal safety and ethical board of the Helmholtz Centre for Infection Research (HZI), (ii) the independent §15 Commission for animal safety (§15 TierSchG), and conducted in accordance with the regulations of the (iii) local government of Lower Saxony (Germany; No. 509.42502-04-017.08). All animals used for experiments in Argentina (e.g., lethal challenge) were cared for in accordance with the Guiding Principles for the care and use of animals of the US National Institute of Health. All procedures were approved by the Institutional Animal Care and Use Committee of the Medical Science School, Universidad Nacional de Cuyo (Protocol approval No. 213/2022).

### 2.4. Influenza Challenge Studies

For challenge studies, the influenza strain A/Puerto Rico/8/34 (H1N1) was used. Virus titers were determined by a focus formation assay as described in Srivastava et al. [65]. Groups of mice (*n* = 6) were shortly anesthetized by intraperitoneal (i.p.) injection of Ketamin-Rompun with a dose adjusted to the individual body weight according to the manufacturer’s instructions (Inresa Arzneimittel GmbH, Freiburg, Germany). Furthermore, mice were challenged on day 60 with a lethal dose of 2 × 10^3^ focus forming units (ffu), or a sublethal dose of 10^3^ ffu of the mouse-adapted H1N1 influenza strain A/Puerto Rico/8/34. Mortality and clinical symptoms of illness, such as weight loss, ruffled fur, hunched posture, and lethargy, were monitored for 14 days after the challenge. Animals showing a weight loss of up to 20% for a sublethal challenge and 25% for a lethal challenge were immediately killed painlessly by slow flooding with CO_2_.

### 2.5. ELISA

NP-specific IgG, IgG1, IgG2a, and IgG2b antibody titers were analyzed in sera from mice collected on days 0, 21 and 42 or 60. Sera were separated from whole blood by centrifugation at 3000× *g* for 10 min and stored at −20 °C. For the measurement of the NP-specific IgA antibody titers in mucosa, animals were sacrificed on day 42, and nasal (NL) and broncho-alveolar lavages (BAL) were obtained by flushing the specific organs with 1 mL of PBS supplemented with 50 mM EDTA, 0.1% bovine serum albumin (BSA) and 10 mM of phenylmethanesulphonyl fluoride (PMSF, Sigma-Aldrich Chemie GmbH, Taufkirchen, Germany). Debris and flushes were eliminated by centrifugation for 10 min at 3000× *g*, supernatant fluids were collected and stored at −20 °C until processing. To perform ELISA assays, plates were coated with 100 μL of rNP [2 μg/mL] and incubated overnight at 4 °C. Twenty-four hours later, plates were blocked with 1% BSA in PBS for 1 h at 37 °C to avoid unspecific binding. Next, serial two-fold dilutions of sera in 1% BSA-PBS were added to the plates (100 μL/well) and incubated for 2 h at 37 °C. After six washes with an auto plate ELISA washer (Biotek ELX405RS, Friedrichshall, Germany) using PBS–0.1% Tween 20, plates were incubated for 2 h at 37 °C with the secondary antibody. For the evaluation of IgA in mucosal lavage samples, we used a biotinylated chain-specific goat anti-mouse IgA (Southern Biotech, Birmingham, AL, USA), whereas for the evaluation of IgG antibodies in serum, biotinylated chain-specific goat anti-mouse IgG (Sigma) or biotinylated rat anti-mouse IgG1, IgG2a, and IgG2b (BD Pharmingen (Heidelberg, Germany) were used. After six washes, 100 μL/well of peroxidase-conjugated streptavidin (BD Pharmingen, Heidelberg, Germany) was added to each well, and plates were incubated at room temperature for 1 h. After another six washes, the detection was performed with the ABTS [2,2-azinobis(3-ethylbenzthiazoline-6-sulfonic acid)] substrate in a 0.1 M citrate-phosphate buffer (pH 4.35) containing 0.05% H_2_O_2_. The ELISA endpoint titers were shown as the reciprocal of the highest sample dilution that yielded an optical density (OD) that was 2-times above the mean value of the blank.

Measurement of TGFβ levels in supernatants of antigen-restimulated splenocyte cultures was performed using the Human/Mouse TGF beta 1 ELISA Ready-SET-Go! kit (2nd Generation/eBioscience/Thermofisher (Waltham, MA, USA)) according to the manufacturer’s instructions.

### 2.6. ELISPOT

Spleens from vaccinated mice were harvested and disaggregated using cell strainers. To lysate red blood cells, the pellet was resuspended in ammonium-chloride-potassium (ACK) lysis buffer. After washing, splenocytes were resuspended in complete RPMI, and the cell number was determined using a Z2 cell counter (Beckman Coulter GmbH, Krefeld, Germany). For IFN-γ, IL-2, IL-17, and IL-4 ELISPOTs, splenocytes were seeded in culture plates in triplicates (1 × 10^6^ or 5 × 10^5^ cells/well) and incubated in the absence or presence of 2 μg/mL of rNP, 24 h for IFN-γ or 48 h for IL-2, IL-17 and IL-4, at 37 °C and 5% CO_2_. After 24 h or 48 h IFN-γ, IL-2, IL-17 and IL-4 ELISPOT kits (BD Pharmingen, San Diego, CA, USA) were used according to the manufacturer’s instructions. The plates were analyzed using an ELISPOT reader and ImmunoSpot image analyzer software v3.2 (CTL, Cleveland, OH, USA).

### 2.7. Proliferation Assay

For the proliferation assay, splenocytes from vaccinated groups (5 × 10^5^ cells/well) were incubated for 96 h in the presence of the indicated concentrations of rNP. After 72 h, 1 µCi of [^3^H] thymidine (Amersham Biosciences Europe GmbH, Freiburg, Germany) was added to each well. After 16–18 h of incubation, cells were harvested on filter mat A (Wallac /PerkinElmer, Rodgau, Germany) using a cell harvester (Inotech, Au, Switzerland), and the [^3^H] thymidine uptake into the DNA of proliferating cells was determined using a scintillation counter (Wallac 1450 Micro-Trilux/PerkinElmer, Rodgau, Germany). The results are presented as stimulation index (SI), which is represented by the counts per minute (cpm) of antigen-stimulated samples (0.1, 1, 2, and 4 µg/mL) divided by the cpm of unstimulated samples (0 µg/mL).

### 2.8. Flow Cytometric Analysis of Multifunctional T Cells and Cytokine Profiling

The capacity of the NP vaccine formulations to stimulate antigen-specific multifunctional CD4^+^ and CD8^+^ T cells producing different T helper cytokines was evaluated by flow cytometry as described previously [66,67]. In brief, splenocytes (2 × 10^7^ cells per well) were incubated (37 °C, 5% CO_2_) in RPMI containing 5 µg/mL of rNP or without antigen to determine the basal cytokine production. After 16–20 h, 5 µg/mL brefeldin A (Sigma-Aldrich Chemie GmbH, Taufkirchen, Germany) was added, and cells were further incubated for an additional 6 h. Subsequently, immune cells were stained for CD3, CD4, and CD8 surface markers (BD Pharmingen, Heidelberg, Germany); CD4, eBioscience/Thermofisher (Waltham, MA, USA) and dead cell markers (Fixable Dead Cell Stain, Invitrogen, Waltham, MA, USA). Furthermore, cells were fixed using 2% (*w*/*v*) p-formaldehyde (PFA), permeabilized for 60 min on ice using 0.5% (*w*/*v*) Saponin in PBS/0.5% (*w*/*v*) BSA and finally stained for intracellular cytokines (i.e., IL-2, IFN-γ (BD Pharmingen, Heidelberg, Germany), IL-17, IL-4 and TNF-α (eBioscience / Thermofisher, Waltham, USA). After 30 min of incubation, stained immune cells were washed twice using PBS and resuspended in PBS for FACS analysis using the BD^TM^ LSRII flow cytometer (BD Pharmingen, Heidelberg, Germany). After spectral overlap compensation with the BD FACS Diva software, data were analyzed using FlowJo v10 (Tree Star, Ashland, OR, USA) on the basis of the following gating strategy: viable singlet leukocytes were gated for CD3^+^, CD4^+^, CD8^+^ and subsequently analyzed for the expression of IL-2, IL-4, IL-17, TNF-α and IFN-γ.

### 2.9. Multiplex FlowCytomix (Cytometric Bead Array)

Supernatants of antigen-restimulated splenocytes have been used to characterize the cytokine profiles using the Th1/Th2/Th9/Th17 FlowCytomix immunoassay from Biolegend (San Diego, CA, USA) according to the manufacturer’s instructions.

### 2.10. Statistical Analysis

Data from each experiment are presented as mean values ± SEM. Statistical analysis between the adjuvanted and non-adjuvanted groups was performed by two-tailed Student’s *t*-test and a two-way ANOVA. Values of *p* < 0.05 (*), *p* < 0.05 (**), *p* < 0.001 (***), *p* < 0.0001 (****) were considered statistically significant. All statistical analyses were performed using the GraphPad Prism 8 software (GraphPad Software, Boston, MA, USA).

## 3. Results

### 3.1. Mucosal Administration of rNP with BPPcysMPEG Stimulates Strong NP-Specific Systemic and Local Humoral Immune Responses

Previous studies suggested that anti-NP IgG antibodies contribute to enhancing cellular immune responses and protecting against influenza [33,34,68]. In order to evaluate NP-specific systemic humoral immune responses, anti-NP IgG antibodies titers were measured in the sera of mice vaccinated by either i.n. or s.c. routes. At day 42, mice vaccinated with two doses of rNP with BPPcysMPEG, indicated a significant increment of anti-NP IgG antibody titers (**** *p* < 0.0001 and ** *p* < 0.01) by both routes, compared with the non-adjuvanted formulation (Figure 1A,B).

Remarkably, while the subcutaneous administration of rNP with BPPcysMPEG resulted in about a 3-fold increase of NP-specific antibody titers compared with the non-adjuvanted formulation, anti-NP IgG titers were more than 100-fold increased after i.n. administration of the same formulation (Figure 1A,C). In addition, the administration of rNP plus BPPcysMPEG by s.c. route did not substantially modulate the anti-NP IgG1, IgG2a, and IgG2b antibody subtype ratio (IgG1/IgG2a ratio 5.3) compared with the one stimulated by rNP alone, (IgG1/IgG2a ratio 9.5). The mucosal application of rNP co-administered with BPPcysMPEG by the i.n. route resulted in a IgG1 > IgG2a > IgG2b dominated response with higher titers (>2 × 10^5^) and a IgG1/IgG2a ratio of 2.4. In contrast, the mucosal administration of rNP alone resulted in low IgG subclass titers with a balanced IgG1/IgG2a ratio of 0.75 (Figure 1B,D).

IgA mucosal immunity plays an important role in protecting against airborne viruses, and it is mainly induced by mucosal vaccination [43,69,70]. In order to evaluate local humoral immune responses, anti-NP IgA antibodies were measured in respiratory mucosal lavages of mice vaccinated intranasally. It was found a significantly increment of anti-NP IgA titer in NL (* *p* < 0.1) and BAL (** *p* < 0.01), compared with the non-adjuvanted rNP formulation (Figure 2A,B).

### 3.2. Mucosal Administration of rNP with BPPcysMPEG Promotes Strong Th1/Th2/Th17 Cellular Immune Responses

Previously, Prajeeth et al., described that BPPcysMPEG was able to trigger co-stimulatory signals to induce a more efficient antigen presentation and, thereby, enhanced T cell priming when co-administered with soluble antigens in mice [61]. In order to analyze and characterize cellular immune responses in mice immunized with rNP plus BPPcysMPEG, antigen-specific lymphoproliferative ELISPOT assays and Th1/Th2/Th9/Th17 FlowCytomix immunoassays were performed.

Mice vaccinated with rNP plus BPPcysMPEG intranasally showed significant splenocyte proliferative capacity (**** *p* < 0.0001) following ex vivo restimulation with increasing concentrations of rNP, compared with the non-adjuvanted formulation (Figure 3A). In contrast, mice vaccinated with the adjuvanted formulation subcutaneously did not show a significant difference in the proliferative capacity after ex vivo restimulation with increasing rNP concentrations (Figure 3B).

The number of cells producing the cytokines IFNγ-, IL-2-, IL-4-, and IL-17- was evaluated by ELISPOT assays three weeks after boosting. High numbers of IL-17- (**** *p* < 0.0001), followed by IL-2- (**** *p* < 0.0001), IFNγ- (**** *p* < 0.0001), and IL-4-secreting cells (**** *p* < 0.0001), were shown in mice vaccinated with rNP plus BPPcysMPEG intranasally, compared with the non-adjuvanted formulation (Figure 3C). On the other hand, a significant increase in the number of IL-4- secreting cells (**** *p* < 0.0001), followed by IL-2- secreting cells (**** *p* < 0.0001) was observed in mice vaccinated with the adjuvanted formulation subcutaneously. Nevertheless, the numbers of IFNγ-secreting cells were almost equivalent to those of the non-adjuvanted formulation (Figure 3D).

FlowCytomix analysis showed that splenocytes from mice vaccinated with the adjuvanted formulation intranasally produced mainly significant levels of Th1 cytokines as IFNγ (**** *p* < 0.0001) and IL-2 (* *p* < 0.1), and Th17 cytokines as IL-17F (** *p* < 0.01), whereas the non-adjuvanted formulation stimulated mainly IFNγ and IL-2 (Figure 4A and Appendix AA). On the other hand, mice vaccinated with the adjuvanted formulation subcutaneously produced mainly IL-2 (** *p* <0.01), compared with the non-adjuvant formulation (Figure 4B and Appendix AB).

These results were indicative that i.n. administration of rNP plus BPPcysMPEG induced a strong Th1/Th2/Th17 immune response, whereas s.c. administration induced a weaker Th1/Th2 immune response. Therefore, co-administration of BPPcysMPEG to rNP by the s.c. route did not substantially improve the cellular immune response that had already been mounted by immunization with rNP alone.

### 3.3. Mucosal Vaccination of Mice with rNP Co-Administered with BPPcysMPEG Enhances the Quality of Antigen-Specific Cellular Response by Stimulating Multifunctional CD4^+^ T Cells

There is previous evidence that multifunctional T cells were associated with enhanced protection against certain infections [71,72,73]. Consequently, the quality of the NP-specific T cell responses was evaluated by their capacity to produce several cytokines by intracellular Flow cytometry. As can be seen in the vaccination, rNP co-administered with BPPcysMPEG by the i.n. route efficiently stimulated multifunctional CD4^+^ T cells as indicated by 9% of bi-functional (IFNγ+/TNFα+, IFNγ+/IL-2+, TNFα+/IL-2+) and 11% of trifunctional (IFNγ+/TNFα+/IL-2+) antigen-specific CD4^+^ T cells, whereas the non-adjuvanted formulation was ineffective to stimulate trifunctional CD4^+^ T cells (Appendix AA). In contrast, mice immunized by the s.c. route showed no significant differences in the amount of multifunctional CD4^+^ T cells in both the adjuvanted and non-adjuvanted groups, with around 10% of bi- and trifunctional cells (Appendix AB).

However, only mice immunized with rNP alone by the s.c. route showed significant NP-specific IFNγ+-producing CD8^+^ T cells (Appendix AB). This is in line with the levels of transforming growth factor-beta (TGF-β) obtained following vaccination with rNP alone and rNP co-administered with BPPcysMPEG by both immunization routes. TGFβ is a crucial regulator of T cell responses. It plays a vital role in regulating responses of innate and adaptive immune cells, e.g., the downregulation of effector functions of CD8^+^ cytotoxic T cells (CTLs) [74]. Thus, mice vaccinated with the adjuvanted and non-adjuvanted formulations by the i.n. route showed substantial concentrations of TGFβ, while mice vaccinated by the s.c. route showed a higher TGFβ titer only when the adjuvanted formulation was used (Appendix AA,B).

### 3.4. Mucosal Vaccination of Mice with rNP plus BPPcysMPEG Confers Protection against Influenza Infection

We next determined whether the immune responses induced by the formulations were able to provide protection against influenza infection. Body weight loss as well as symptoms of infection, such as huddling, ruffled fur, and lethargy, were monitored daily for at least two weeks.

In a first attempt, mice were challenged on day 60 with a sub-lethal dose of the mouse-adapted H1N1 influenza strain A/Puerto Rico/8/34. Mice vaccinated with BPPcysMPEG by the i.n. route showed enhanced protection with almost no influence on the loss of weight after sub-lethal challenge with homologous influenza strain A/Puerto Rico/8/34 (H1N1) and were statistically significant (***, *p* < 0.001) compared with rNP alone on day 7 (Figure 5C). Notably, the efficacy of the formulation encompassing rNP alone was increased after s.c. application compared with the i.n. vaccination strategy, highlighting the necessity of an adjuvant for i.n. vaccination approaches. Mice vaccinated with adjuvanted rNP formulation by the s.c. route showed a stronger protection level than mice vaccinated with rNP alone; however, differences weren’t statistically significant compared with rNP alone (Figure 5D). It was observed, that mice vaccinated with rNP plus BPPcysMPEG by the i.n. route showed no influence on the weight after sub-lethal challenge with homologous influenza strain A/Puerto Rico/8/34 (H1N1). Moreover, animals vaccinated with rNP co-ad- ministered with BPPcysMPEG by the i.n. route showed reduced morbidity and statistically relevant (***, *p* < 0.001) gradual weight with only mild weight loss and recovery after 7 days post-infection. Subsequently, mice vaccinated with rNP plus BPPcysMPEG intranasally were challenged with a lethal dose of influenza strain A/Puerto Rico/8/34 (H1N1).

It was observed that mice vaccinated with rNP co-administered with BPPcysMPEG were protected and recovered their weight within 10 days, whereas all mice vaccinated with rNP alone as well as control mice showed severe symptomatology and high mortality between the 6th and 8th day (Figure 5A,B). Animals showing a weight loss of up to 25% at day 7 were immediately killed painlessly by slow flooding with CO_2_ (Figure 5A,B).

## 4. Discussion

Current influenza vaccines are designed to target specific strains and therefore must be updated annually. In addition, they mostly induce immune responses mediated by antibodies specific for the surface glycoproteins but are incapable of inducing strong cellular immune responses, resulting in reduced vaccine efficacy [75]. It is assumed that the induction of immunity requires balanced humoral and cell-mediated immunity. Most current vaccines lack the ability to induce robust cytotoxic and long-term immunity, as well as heterosubtypic responses [76,77]. Vaccines formulated with conserved antigens could be the key inducing robust cellular immune responses able to protect against antigenic variants of influenza [78]. Mathematical models showed that if just 10 percent of current influenza vaccinations were replaced by a “universal” flu vaccine, approximately 5.3 million fewer people would be infected. Moreover, approximately 6000 fewer people would die each year in the US, and there would be over one billion dollars saved in healthcare expenses [79]. Furthermore, it has been demonstrated previously that vaccination against influenza by the i.n. route has considerable benefits over the conventional i.m. administration route [80]. Thus, besides stimulating a more appropriate immune response that mimics a natural infection, i.n. vaccination provides a longer-lasting effect and is characterized by the ease of administration and increased acceptance by the public [81]. Therefore, the aim of this study was to evaluate an experimental mucosal influenza vaccine based on recombinant NP formulated with a diacylated lipopeptide ligand of the TLR2/6 heterodimer, the BPPcysMPEG. Nucleoprotein was selected as a candidate antigen since it has been demonstrated that it is able to induce lymphocyte-mediated protection against different influenza virus strains [82,83,84]. Numerous examples of influenza vaccines based on viral vectors, which include NP, have been successful in inducing very strong cellular and humoral immune responses resulting in heterologous protection in pre-clinical models and clinical trials [26,27,85,86,87]. Nevertheless, viral vector-based vaccines are cumbersome and costly to produce, and many of them have several concerns related to pre-existing vector immunity.

In this study, we proposed a subunit vaccine that contains the recombinant nucleoprotein; however, it is well known that recombinant proteins are generally poor immunogens, especially when administered by the i.n. route [80]. Therefore, we planned the addition of an adjuvant in order to enhance its immunogenicity. BPPcysMPEG is a compound with a specific recognition and signaling pathway through TLR2 via heterodimerization with TLR6. BPPcysMPEG is a synthetic analogue of MALP-2, whose adjuvanticity has been improved by enhancing its bioavailability through pegylation [61]. Previous studies have shown that MALP-2 acts as a strong adjuvant by promoting DC activation and maturation as well as modulating their protein processing pattern from proteasomes to immunoproteasomes, showing increased proteolytic activity and antigen presentation [52,58,61,88].

Our results demonstrated that the addition of BPPcysMPEG to rNP induced significant improvements in antigen-specific humoral and cellular responses. The results underscored the activity of BPPcysMPEG as an adjuvant, as indicated by the remarkable high serum IgG titers elicited after i.n. and s.c. immunization compared with those obtained when immunizing with rNP alone. Interestingly, it was observed that only when rNP was co-administered with BPPcysMPEG intranasally, NP-specific IgG2b subclass titer was increased. Similar to IgG1 and IgG2a, IgG2b has also been shown to be involved in protection against influenza infection [89]. Thus, while IgG1 is characterized by its neutralizing function, IgG2a and IgG2b initiate the antibody-dependent cellular cytotoxicity (ADCC) [90]. Although the described effects mainly focus on antibodies specific for influenza HA and NA, there is growing evidence that NP-specific IgG antibodies also play a significant role in animal and human influenza protection [33,34,91]. These antibodies form antigen-antibody complexes that are recognized by dendritic cells through Fc receptors. This phenomenon triggers subsequent antiviral reactions that cooperate with CD8^+^ T cells to eliminate virally infected cells [33,34]. In addition, it has been demonstrated in humans that anti-NP IgG also promoted ADCC contributing to heterosubtypic protection [32]. Probably one of the most relevant features of BPPcysMPEG shown in this work is its ability to induce significant NP-specific IgA antibody secretion in the mucosal territories of the lung and nose. While an influenza-specific IgG titer in the sera prevents the disease and to some extent limits viral lung pathology, IgA titers are needed to eliminate nasal viral shedding, thereby reducing viral transmission rates [92]. Moreover, in contrast to influenza-specific IgG, IgA seems to be highly cross-reactive and protects against infection by both homologous and heterologous viruses [10]. Based on the present results, it is likely that these anti-NP IgA antibodies in mucosal territories have contributed to the protection against challenge. In fact, other works have reported protection in mice against influenza virus challenge partially mediated by anti-NP specific IgA [70,93]. Regarding the significance of the IgA antibody, other viral infection models have demonstrated that IgA directed against other internal proteins of the virion may have access to the infected cell interior and neutralize the virus by interfering with the mechanism of replication [94,95]. The mechanism of this phenomenon postulates that extracellular virus-specific antibodies can get access to the cytosol of infected cells through the process of transcytosis, and we think that a similar mechanism of transcytosis might be considered in our work.

Recently, it has been shown that cellular responses against different influenza antigens strongly enhance vaccine performance. Since several studies have demonstrated previously that BPPcysMPEG is able to enhance cellular immune responses against different antigens [57,58,60,88], we analyzed the potential of the NP vaccine to stimulate cellular immunity. When rNP was administered together with BPPcysMPEG intranasally, not only was the proliferative capacity of splenocytes significantly increased, but there was also an increase in cytokine responses with a mixed Th1/Th2/Th17 response. In line with previous studies showing that mucosal vaccination promotes the stimulation of Th17 responses [58,96,97,98], the i.n. administration of rNP with BPPcysMPEG stimulated strong NP-specific IL-17 responses. Interleukin-17 acts as a potent pro-inflammatory cytokine and plays a crucial role in pulmonary host defense against diverse pathogens [98,99,100]. However, the role of IL-17 during influenza infection is not yet fully defined. For example, IL-17-secreting CD4^+^ T and CD8^+^ T effector cells seem to have a protective role in the lung of mice following a primary challenge with influenza A, whereas neutralization of IL-17 abrogates this effect [101]. In addition, H5N1-infected IL-17 knockout mice exhibit increased morbidity and mortality when compared with infected wildtype controls [102].

In contrast, Gopal et al., demonstrated that mucosal pre-exposure to Th17-inducing adjuvants resulted in increased morbidity and exacerbated lung inflammation upon subsequent infection with different influenza A strains [103]. However, this potentially negative effect of our vaccine formulation might be compensated by the increased levels of the regulatory T cell effector cytokine TGFβ, which we observed in vaccinated mice. Egarnes and Gosselin demonstrated that increased levels of TGFβ correlated with a significant decrease in Th17 cells and an anti-inflammatory environment, allowing better control of inflammation in influenza-infected patients [104]. In addition, TGFβ also controls the function of CD4^+^ and CD8^+^ T cells during influenza infection, which could also explain the marginal CD8^+^ responses obtained after immunization of mice with rNP plus BPPcysMPEG [105]. Nevertheless, incorporation of BPPcysMPEG in the intranasal vaccine not only increased the strength but also the quality of the NP-specific CD4^+^ T cell response, as indicated by the increment of bi- and trifunctional CD4^+^ T cells. It has already been demonstrated that multifunctional CD4^+^ T cells contribute to protection against influenza infections [106]. For example, Trieu and co-workers revealed that annual vaccination results in a booster of multifunctional memory CD4^+^ T cells following each vaccination. Moreover, incorporation of the same influenza antigen in subsequent annual vaccines can significantly impact the influenza immune response [107].

The addition of BPPcysMPEG to the rNP formulation improved the efficacy of protection compared with vaccination with rNP alone when given by the i.n. route, as well as by the s.c. route, as demonstrated in the sublethal challenge. However, the lower efficacy of the rNP alone formulation shown in the i.n. vaccination strategy highlighted the power of BPPcysMPEG to enhance mucosal protection. This was also verified after the lethal challenge, where all animals were protected. Although the sublethal challenge dose-based infection model showed a clear trend toward effective protection due to the s.c. formulation compared with rNP alone, the protective efficacy against lethal infection remains elusive. Thus, even though the infection model based on a sublethal dose resulted in robust efficacy data, it would be important to perform a lethal challenge in a future experiment to measure the protective efficacy of the s.c. formulation. The immune responses induced by the BPPcysMPEG i.n. formulation were consistent with the improved protection efficacy. Mice vaccinated intranasally showed improved protection against sub-lethal and lethal challenges with the homosubtypic H1N1 strain as compared with those vaccinated with rNP alone. The same seems to be true when mice were immunized by the s.c. route. However, the protective performance of the s.c. formulation against lethal infection remains elusive. Thus, although the infection model based on a sub-lethal dose resulted in robust efficacy data, it would be important to perform a lethal challenge in a future experiment to measure the protective efficacy of the s.c. formulation. In this regard and in view of the goal of developing a universal influenza vaccine, further experimentation and efforts are needed to make our rNP + BPPcysMPEG formulation a candidate for a universal vaccine. Therefore, further studies need to be performed to investigate whether our vaccine candidate is able to stimulate protective immune responses against not only homo- but also hetero-subtypic influenza strains. To this end, we are aiming to perform immunogenicity and efficacy studies in the highly accepted influenza ferret model.

Regarding the NP antigen used, several clinical trials of NP-containing vaccines have been conducted, and although the use of NP antigen in influenza vaccines is not yet available, it is possible that this antigen will be a strong candidate in universal vaccines in the near future [108,109]. Another point to bear in mind is that there are few mucosal vaccines on the market, and one of the reasons for this limitation is the lack of effective and safe mucosal adjuvants. Since the COVID-19 pandemics, progress in the use and approval of new vaccines has accelerated dramatically, and there is growing confidence that new mucosal vaccines could be introduced into human vaccines [110]. In sum, this work clearly demonstrates that BPPcysMPEG is a promising adjuvant that could be considered for the development of innovative mucosal vaccines against respiratory pathogens, such as influenza.

## 5. Patent

CAG is named as an inventor in a granted patent covering the use of BPPcysMPEG as an adjuvant (PCT/EP 02007640).

## Figures and Tables

**Figure 1 pharmaceutics-15-00912-f001:**
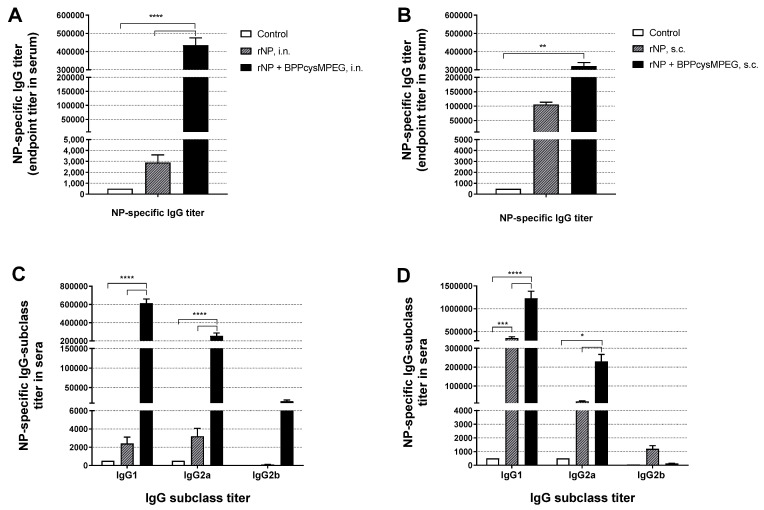
Evaluation of NP-specific IgG and IgG subclasses antibodies in serum of mice vaccinated with the formulations by i.n. and s.c. route. NP-specific IgG titer and IgG subclasses titers (IgG1, IgG2a and IgG2b) observed in mice vaccinated with PBS (control), rNP or rNP co-administered with BPPcysMPEG vaccinated by (**A**,**C**) i.n. or (**B**,**D**) s.c. route, measured by ELISA. The results are expressed as mean endpoint titers with SEM. Statistical analysis between the adjuvanted and non-adjuvanted groups was performed by two-tailed Student’s *t*-test and two-way ANOVA. Differences were statistically significant (* *p* < 0.1), (** *p* < 0.01), (*** *p* < 0.001) and (**** *p* < 0.0001) with respect to values obtained in control mice and/or mice receiving rNP alone.

**Figure 2 pharmaceutics-15-00912-f002:**
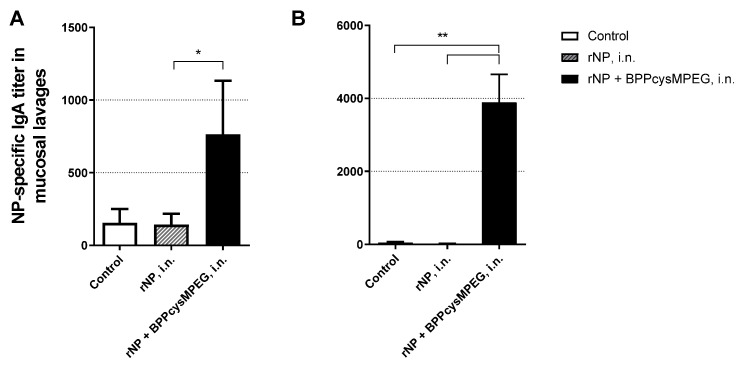
Evaluation of IgA antibody response in mucosal compartments of mice vaccinated with the formulations by i.n. route. NP-specific IgA antibody of (**A**) broncho-alveolar-(BAL) and (**B**) nasal lavages (NL) from mice vaccinated with PBS (control), rNP and rNP co-administered with BPPcysMPEG. The results are expressed as mean endpoint titers with SEM. Statistical analysis between the adjuvanted and non-adjuvanted groups was performed by two-tailed Student’s *t*-test. Differences were statistically significant (* *p* < 0.1) and (** *p* < 0.01) with respect to values obtained in control mice and/or mice receiving rNP alone.

**Figure 3 pharmaceutics-15-00912-f003:**
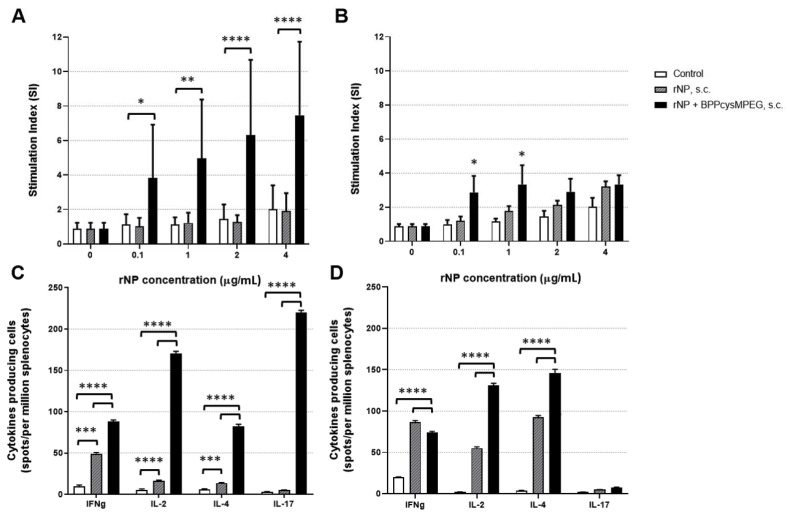
Analysis of the NP-specific cellular responses of mice vaccinated with the formulations by i.n. and s.c. route. Proliferation of splenocytes from mice with PBS (control), rNP and rNP co-administered with BPPcysMPEG, stimulated with increasing concentration of NP measured by the incorporation of the radioactivity of [^3^H] thymidine. Results are expressed as stimulation index (SI), which is the ratio of [^3^H]-thymidine uptake of stimulated versus unstimulated samples (**A**,**B**). Number of IFN-γ, IL-2, IL-4, and IL-17-producing cells were evaluated in splenocytes by ELISPOT. Results are expressed as the number of spots of cytokine-producing cells per 10^6^ spleen cells after subtraction of background values of unstimulated cells (**C**,**D**). Statistical analysis between the adjuvanted and non-adjuvanted groups was performed by two-tailed Student’s *t*-test. Differences were statistically significant (* *p* < 0.1), (** *p* < 0.01), (*** *p* < 0.001) and (**** *p* < 0.0001) with respect to values obtained in control mice and/or mice receiving rNP alone.

**Figure 4 pharmaceutics-15-00912-f004:**
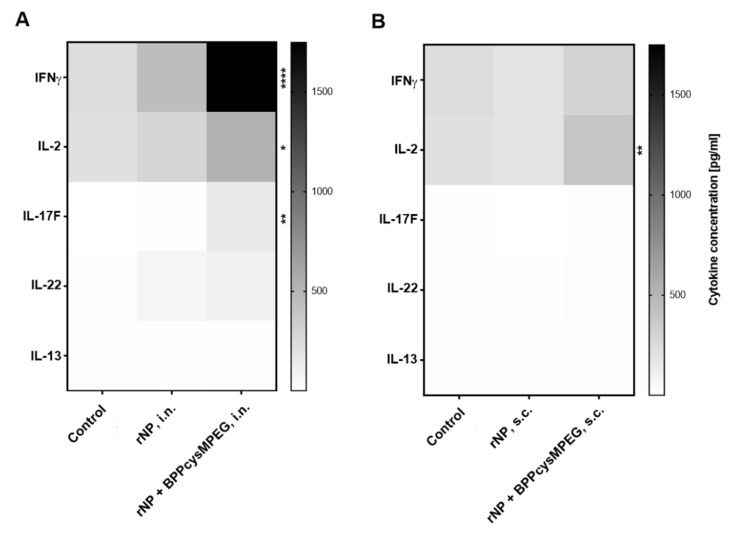
Cytokine profiles stimulated by rNP co-administered with BPPcysMPEG. The presence of mouse IFN-γ, IL-2, IL-13, IL-17F and IL-22 were determined using a cytometric bead array. Results are presented as a heat map of Th1, Th2 and Th17 cytokines secreted by antigen-restimulated splenocytes derived from (**A**) mice vaccinated by i.n. and (**B**) by s.c. route. Statistical analysis between the adjuvanted and non-adjuvanted groups was performed by two-tailed Student’s *t*-test and two-way ANOVA. Differences were statistically significant (*, *p* < 0.1; **, *p* < 0.01; ****, *p* < 0.0001), compared to control mice.

**Figure 5 pharmaceutics-15-00912-f005:**
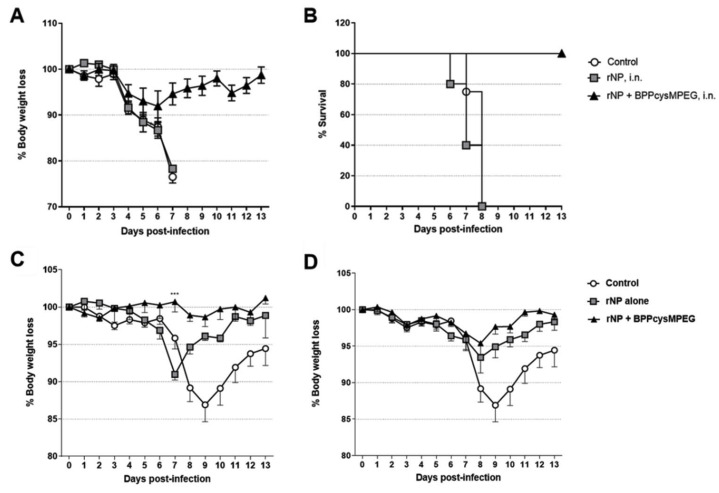
Protection of mice vaccinated with the formulations against influenza A virus infection. Vaccinated BALB/c mice groups were challenged with a lethal dose of the homologous influenza strain A/Puerto Rico/8/34 (H1N1) on day 60. Bodyweight loss in percentage (**A**) and survival rates (**B**) were measured after lethal challenge for a period of two weeks. SEM are indicated by vertical lines. Vaccinated BALB/c mice groups were challenged with a sublethal dose (10^3^ ffu) of the homosubtypic influenza strain A/Puerto Rico/8/34 (H1N1) on day 60 by the i.n. (**C**) or the s.c. (**D**) route. Bodyweight loss was measured daily after challenge for a period of two weeks. SEM are indicated by vertical lines. Statistical analysis between the adjuvanted and non-adjuvanted groups was performed by two-tailed Student’s *t*-test. Differences were statistically significant (***, *p* < 0.001) compared to NP.

## Data Availability

The datasets generated during and/or analyzed during the current study are available from the corresponding author on reasonable request.

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
