# Peer review of "Protective Efficacy of a Mucosal Influenza Vaccine Formulation Based on the Recombinant Nucleoprotein Co-Administered with a TLR2/6 Agonist BPPcysMPEG"

_pharmaceutics, 2023, doi:10.3390/pharmaceutics15030912_

Round 1

Reviewer 1 Report

The manuscript is very well written and the result are largely presented. The discussion is also relevant to the content of the paper.

Some English suggestions are attached to the file.

Author Response

Reviewer 1:

The manuscript is very well written and the result are largely presented. The discussion is also relevant to the content of the paper.

Some English suggestions are attached to the file.

Answer: We appreciate the reviewer´s comment, and the text was modified, adapted as required and marked in yellow.

Reviewer 2 Report

This manuscript reports on the immune response to vaccination with a recombinant subunit vaccine based on influenza nucleoprotein in mice when administered intranasally or by subcutaneous injection, with or without the TLR2/6 agonist BPPcysMPEG as adjuvant.

The main findings are:

·       When administered i.n. the adjuvant is required but not when administered s.c.

·       The nature of the immune response varies dependent on the route of administration.

·       An adjuvanted vaccine administered i.n. is effective in providing protection in lethal challenge studies.

The manuscript is very clearly written, and the studies well carried out and described.

However, I have several reservations which lead me to recommend that the manuscript is not suitable for publication in Pharmaceutics in the current form.

Specifically:

·       There is no information on the actual antigen beyond it being a recombinant subunit produced in bacteria. – Where is it from. How is it expressed and purified. What is the level of purity and how is this determined. How is structural integrity assessed?

·       The challenge studies used have a major discrepancy in that the data shown for the i.n. administered vaccine uses a lethal dose while that for the s.c route is a sublethal dose. I believe that this makes a direct comparison impossible. Yes, in the case of the i.n. adjuvant is necessary for protection. But how do we know that that the s.c. administered vaccine without adjuvant is not capable of providing protection against a lethal dose? To my mind this is “an elephant in the room” which the manuscript in current form rather glosses over.

Minor points that need to be addressed before publication:

·       Figures 1 and 3 need to be reproduced at a larger scale to make them legible.

·       The axis in panels B and D of figure 3 needs to be labelled.

·       The greyscale used in Figure 4 makes the differences at lower levels hard to determine, perhaps using colour would help here.

·       The 7th line of the Discussion is missing a reference.

Author Response

Reviewer 2:

This manuscript reports on the immune response to vaccination with a recombinant subunit vaccine based on influenza nucleoprotein in mice when administered intranasally or by subcutaneous injection, with or without the TLR2/6 agonist BPPcysMPEG as adjuvant.

The main findings are:

  • When administered i.n. the adjuvant is required but not when administered s.c.
  • The nature of the immune response varies dependent on the route of administration.
  • An adjuvanted vaccine administered i.n. is effective in providing protection in lethal challenge studies.
  • The manuscript is very clearly written, and the studies well carried out and described.
  • However, I have several reservations which lead me to recommend that the manuscript is not suitable for publication in Pharmaceutics in the current form.

We appreciate the comments of reviewer 2 and have corrected the manuscript to address reviewer criticisms.

Specifically:

  1. There is no information on the actual antigen beyond it being a recombinant subunit produced in bacteria. – Where is it from? How is it expressed and purified? What is the level of purity and how is this determined? How is structural integrity assessed?

The nucleoprotein gene derived of influenza strain A/PR/8/34 (H1N1) was cloned into the pET30a plasmid and the protein expressed in Escherichia coli BL21, (DE3) bacteria. The protein was used in previous works of Cargnelutti et al. 1–3. The expression, purification, and characterization of rNP Protein is descripted in detail in previous work 1. We incorporated the references to the manuscript.

“For vaccination studies, groups of 5 mice were immunized with 10 μg/dose rNP derived from the influenza strain A/PR/8/34 (H1N1) alone or combined with 10 μg/dose BPPcysMPEG (HZI research-grade quality), administered by the i.n. (20 μl) or s.c. (100 µl) route on days 0 and 21.” Line 162-165

References

  1. Cargnelutti DE, Sanchez M V, Alvarez P, et al. Improved immune response to recombinant influenza nucleoprotein formulated with ISCOMATRIX. J Microbiol Biotechnol. 2012;22(3):416-421. http://www.ncbi.nlm.nih.gov/pubmed/22450799
  2. Cargnelutti DE, Sanchez MV, Alvarez P, Boado L, Mattion N, Scodeller EA. Enhancement of Th1 immune responses to recombinant influenza nucleoprotein by Ribi adjuvant. New Microbiol. 2013;36(2).
  3. Sanchez MV, Ebensen T, Schulze K, et al. Intranasal delivery of influenza rNP adjuvanted with c-di-AMP induces strong humoral and cellular immune responses and provides protection against virus challenge. PLoS One. 2014;9(8). doi:10.1371/journal.pone.0104824
  4. The challenge studies used have a major discrepancy in that the data shown for the i.n. administered vaccine uses a lethal dose while that for the s.c route is a sublethal dose. I believe that this makes a direct comparison impossible. Yes, in the case of the i.n. adjuvant is necessary for protection. But how do we know that that the s.c. administered vaccine without adjuvant is not capable of providing protection against a lethal dose? To my mind this is “an elephant in the room” which the manuscript in current form rather glosses over.

Reviewer 2 raises a pertinent concern. We agree with the reviewer, that it would have been more appropriate to perform lethal challenge for subcutaneous immunization as well.

In fact, both studies – i.n. and s.c. application– have been performed using a sub-lethal dose. We believe that our model based on up to 20% of weight loss of mice receiving only PBS or in case of vaccine failure is strong enough to allow robust conclusions about the efficacy of our vaccine candidates.

Moreover, the animal studies performed are in accordance with animal ethics based on the international principles of the 3Rs (Replacement, Reduction and Refinement), aiming in the avoidence of unnecessary animal suffering. In this regard, due to new animal law did not allow now to perform lethal dose challenge studies.

Finally, with the present work we didn´t aim in showing that the i.n. route being superior to the s.c. route. We rather wanted to demonstrate that our BPPcysMPEG adjuvanted vaccine candidate is stimulating protective responses by both, parenteral and mucosal routes. Thus, while the i.n. strategy has some advantages compared to the s.c. strategy (e.g. superior local responses able to block already infection, but also transmission), the latter reflects more the classical parenteral vaccination strategy which - thinking ahead - might get easier permission for human use.

We consider to include the figure and the results of the i.n. described above in the supplementary Fig. 3 A/B. We modified the text of results and have considered pertinent to include that it was not possible to establish whether subcutaneous immunization would be able to protect against lethal challenge, and that it should be done in a future experiment.

We rephrased the corresponding paragraph. “Firstly, mice were challenged on day 60 with a sub-lethal dose of the mouse-adapted H1N1 influenza strain A/Puerto Rico/8/34. Mice vaccinated with rNP plus BPPcysMPEG intranasally, showed a better protective efficacy than mice vaccinated with rNP plus BPPcysMPEG by the subcutaneous route, noting that mice vaccinated subcutaneously performed similarly to rNP alone (Suppl. Fig. 3 A/B). It was observed, that mice vaccinated with rNP plus BPPcysMPEG by i.n. route showed no influence on the weight after sub-lethal challenge with homologous influenza strain A/Puerto Rico/8/34 (H1N1). Moreover, animals vaccinated with rNP co-administered with BPPcysMPEG by i.n. route showed reduced morbidity and statistical relevant (***, p< 0.001) gradual weight with only weak weight loss and recovery after 7th days post-infection. Subsequently, a lethal challenge with the homologous influenza strain A/Puerto Rico/8/34 (H1N1) of mice vaccinated with rNP co-administered with BPPcysMPEG was performed to confirm the superior efficacy of the mucosal application route.“ Line 400-412

We adapt the text accordingly and include additional information. The superior immune responses induced by BPPcysMPEG i.n. formulation was consistent with the improved protection efficacy. Mice vaccinated intranasally showed improved protection against infection with the homologous H1N1 strain as compared to those vaccinated with rNP alone. The same is true for mice vaccinated subcutaneously with the adjuvanted formulation.“ Line 530-534

Supplementary Figure 3: Protection of mice vaccinated with the formulations by s.c. or i.n. route and subsequent sub-lethal challenge with homologous influenza A (H1N1) virus. Vaccinated BALB/c mice groups were challenged with 103 ffu/dose of the homologous influenza strain A/Puerto Rico/8/34 (H1N1) on day 60 by i.n. (Fig 3 A) or s.c. (Fig. 3 B) route. Bodyweight loss was measured daily after challenge for a period of two weeks. SEM are indicated by vertical lines. Statistical analysis between the adjuvanted and non-adjuvanted groups was performed by two-tailed Student’s t-test. Differences were statistically significant (***, 0.001) compared to NP.

Minor points that need to be addressed before publication

Figures 1 and 3 need to be reproduced at a larger scale to make them legible.

We modified Figure 1 and 3 as required by the reviewer. Line 283 and 331

The axis in panels B and D of Figure 3 needs to be labelled.

We adapt the Figure 3 B/D accordingly and include additional information. Line 331

The greyscale used in Figure 4 makes the differences at lower levels hard to determine, perhaps using colour would help here

We included the additional Supplementary Figure 4 to make the differences in cytokines secretion (in pg/ml) between the analyzed groups more visible for the reader.

Supplementary Figure 4. Cytokine profiles antigen-restimulated splenocytes derived from vaccinated mice. The presence of mouse IFN-γ, IL-2, IL-13, IL-17F and IL-22 were determined using a cytometric bead array. Results are presented as cytokine concentration in [pg/ml] of Th1, Th2 and Th17 cytokines secreted by antigen-restimulated splenocytes derived from (A) mice vaccinated by i.n. and (B) by s.c. route.”

The 7th line of the Discussion is missing a reference.

We appreciate the reviewer´s comments, and the figures were modified as required.

Reviewer 3 Report

The manuscript described the development of a mucosal subunit vaccine adjuvanted with TLR2/6 agonists. Systemic and mucosal immunity was induced and compared with the subunit vaccine without adjuvant.  Moreover, vaccine efficacy was compared with the parenteral vaccination of mice with the same formulation. The topic of the study is interesting and important to the field. However, the experimental design and the analysis/display of the data is quite disappointing. The most critical comparisons between the IN and SC routes of the adjuvanted vaccines were not properly done. It is weird that for the viral challenge study, why lethal dose was chosen for the IN vaccinated mice, while sublethal dose was used for the SC vaccinated one? The comparison of the protect efficacy against challenged virus between the IN and SC routes is crucial.  

The following points are improving the clarity of the manuscript.   

1.     Fig1. The data should be log transferred. 1A and 1B can be combined, and statistical analysis to compare the immune responses in S.C. and I.N. groups can be run. Legends in 1C and 1D are missing.  The authors mentioned “balanced IgG1/IgG2a ratio” for 1CD, please provide the exact number of that ratio.

2.     Fig2. Did the authors measure the IgA responses for the sc route?

3.     Fig 3. Legends in 3C and 3D are missing.  The comparison between responses of IN and SC routes would be interesting.

4.     Fig4. The most interesting comparison would be between responses of IN and SC routes.

5.     In Fig3C, it seems that more IL-17-producing cells were induced compared to IFNg-producing cells. However, in Fig4A, higher IFNg were detected. Please explain this discrepancy.

6.     Page 10, the first motioned “Figure4’ should be “figure 5”.   Add “ref “on page 10.

7.     Fig5. The data on Supplementary fig 3 should put in Fig5.

Author Response

Reviewer 3:

The manuscript described the development of a mucosal subunit vaccine adjuvanted with TLR2/6 agonists. Systemic and mucosal immunity was induced and compared with the subunit vaccine without adjuvant.  Moreover, vaccine efficacy was compared with the parenteral vaccination of mice with the same formulation. The topic of the study is interesting and important to the field. However, the experimental design and the analysis/display of the data is quite disappointing. The most critical comparisons between the IN and SC routes of the adjuvanted vaccines were not properly done. It is weird that for the viral challenge study, why lethal dose was chosen for the IN vaccinated mice, while sublethal dose was used for the SC vaccinated one? The comparison of the protect efficacy against challenged virus between the IN and SC routes is crucial. 

The reviewer raises a pertinent concern. Please, compare our response to the query of reviewer 2 ( point 2)

The following points are improving the clarity of the manuscript.  

  1. Fig 1. The data should be log transferred. 1A and 1B can be combined, and statistical analysis to compare the immune responses in S.C. and I.N. groups can be run.

 Legends in 1C and 1D are missing. 

We appreciate the reviewer´s comments, and the text and Fig. 1 C/D were modified as requested by the reviewer. Line 283

  1. The authors mentioned “balanced IgG1/IgG2a ratio” for 1CD, please provide the exact number of that ratio.

We appreciate the reviewer´s comments, and the text and Fig. 1 CD were modified as required.

Remarkably, while the subcutaneous administration of rNP with BPPcysMPEG resulted in about 3-times increased NP-specific antibody titers compared to the non-adjuvanted formulation, anti-NP IgG titers were more than 100-fold increased after i.n. administration of the same formulation (Fig. 1 A/C). In addition, while the administration of rNP plus BPPcysMPEG by s.c. route did not substantially modulate the anti-NP IgG1, IgG2a and IgG2b antibody subtype ratio (IgG1/IgG2a ratio 5.3) compared to the one stimulated by rNP alone (IgG1/IgG2a ratio 9.5), mucosal application of rNP co-administered with BPPcysMPEG by the i.n. route resulted in a IgG1 > IgG2a > IgG2b dominated response with higher titers (>2 x 105) and a IgG1/IgG2a ratio of 2.4. In contrast, the mucosal administration of rNP alone resulted in low IgG subclass titers with a balanced IgG1/IgG2a ratio of 0.75 (Fig. 1 B/D).” Line 292-302

  1. Fig 2. Did the authors measure the IgA responses for the sc route?

The reviewer raises a pertinent concern.

We did not measure IgA responses stimulated following immunization via the s.c. route because IgA is the dominant antibody at many mucosal sites and most IgA responses are induced using the mucosal route rather than subcutaneous.

In addition. In Fig. 2 A/B, we showed that even the non-adjuvanted rNP formulation given by mucosal application was not able to induce an antigen-specific IgA response at local sites, such as lung (broncho alveolar lavage, BAL) and nasal cavity (nasal lavage, NL).

Our experience in the past, were we tried to detect antigen–specific IgA at local sites in samples derived from mice vaccinated by parenteral (s.c. or i.m.) route in preliminary animal studies. Almost all samples were negative and only a few samples showed a weak non-significant local immune response, thus we did not focus, on this issue. Due to the situation, that we did not store lavage samples from parenteral immunized mice, we will not be able to perform additional ELISA assays.

Nevertheless, we will take care for the pertinent concern of the reviewer and include the storage of lavage samples of parenteral vaccinated animals to be prepared for future questions.

  1. Fig 3. Legends in 3C and 3D are missing. The comparison between responses of IN and SC routes would be interesting.

We appreciate the reviewer´s comments. We rephrased the corresponding paragraph of Fig. 3 as required.

  1. Fig 4. The most interesting comparison would be between responses of IN and SC routes.

We appreciate the reviewer´s comments, and text and Supplementary Figure 4 A/B were modified as required.

Supplementary Figure 4. Cytokine profiles antigen-restimulated splenocytes derived from vaccinated mice. The presence of mouse IFN-γ, IL-2, IL-13, IL-17F and IL-22 were determined using a cytometric bead array. Results are presented as cytokine concentration in [pg/ml] of Th1, Th2 and Th17 cytokines secreted by antigen-restimulated splenocytes derived from (A) mice vaccinated by i.n. and (B) by s.c. route.

  1. In Fig 3C, it seems that more IL-17-producing cells were induced compared to IFNg-producing cells. However, in Fig 4A, higher IFNg were detected. Please explain this discrepancy.

We detected in Fig. 3 C cytokine producing positive cells, whereas in Figure 4 and Supplementary Fig. 4 we analyzed the secreting cytokines of antigen-restimulated splenocytes. Thus, we can´t compare the number of Th17 positive cells (IL-17) and a concentration of cytokines (pg/ml). The same is true for IFNg, or IL-2 postive cells.

In general, we were able to detect both cytokine secreting cells and secreted cytokines in the supernatant of antigen-restimulated splenocytes. We were able to show, that between number of cells and secretion level can be a great range with low number / high secretion versus high number / low secretion. Nevertheless, we observed a strong relation between the stimulated immune cells and did not observe any unspecific secretion by the stimulated cells.

  1. Page 10, the first motioned “Figure 4’ should be “figure 5”. Add “ref “on page 10.

We rephrased the corresponding paragraph. “Mice vaccinated with rNP co-administered with BPPcysMPEG recover their weight within 10 days, whereas all mice vaccinated with rNP alone as well as control mice showed severe symptomatology and high mortality between the 6th and 8th day (Fig. 5 AB). Animals showing a weight loss of up to 25% at day 7 were immediately killed painlessly by slow flooding with CO2 (Fig.5 A/B).” Line 419-423

  1. Fig 5. The data on Supplementary fig 3 should put in Fig 5

We appreciate the reviewer´s suggestion.

We include the sub-lethal challenge of mice vaccinated by i.n. route into the Supplementary Fig. 3. A/B. Now, the reader will be able to compare both application routes as requested by the reviewer.

We rephrased the corresponding paragraph. “Firstly, mice were challenged on day 60 with a sub-lethal dose of the mouse-adapted H1N1 influenza strain A/Puerto Rico/8/34. Mice vaccinated with rNP plus BPPcysMPEG intranasally, showed a better protective efficacy than mice vaccinated with rNP plus BPPcysMPEG by the subcutaneous route, noting that mice vaccinated subcutaneously performed similarly to rNP alone (Suppl. Fig. 3 A/B). It was observed, that mice vaccinated with rNP plus BPPcysMPEG by i.n. route showed no influence on the weight after sub-lethal challenge with homologous influenza strain A/Puerto Rico/8/34 (H1N1). Moreover, animals vaccinated with rNP co-administered with BPPcysMPEG by i.n. route showed reduced morbidity and statistical relevant (***, p< 0.001) gradual weight with only weak weight loss and recovery after 7th days post-infection. Subsequently, a lethal challenge with the homologous influenza strain A/Puerto Rico/8/34 (H1N1) of mice vaccinated with rNP co-administered with BPPcysMPEG was performed to confirm the superior efficacy of the mucosal application route.“ Line 400-412

We adapt the text accordingly and include additional information. The superior immune responses induced by BPPcysMPEG i.n. formulation was consistent with the improved protection efficacy. Mice vaccinated intranasally showed improved protection against infection with the homologous H1N1 strain as compared to those vaccinated with rNP alone. The same is true for mice vaccinated subcutaneously with the adjuvanted formulation.“ Line 530-534

Supplementary Figure 3: Protection of mice vaccinated with the formulations by s.c. or i.n. route and subsequent sub-lethal challenge with homologous influenza A (H1N1) virus. Vaccinated BALB/c mice groups were challenged with 103 ffu/dose of the homologous influenza strain A/Puerto Rico/8/34 (H1N1) on day 60 by i.n. (Fig 3 A) or s.c. (Fig. 3 B) route. Bodyweight loss was measured daily after challenge for a period of two weeks. SEM are indicated by vertical lines. Statistical analysis between the adjuvanted and non-adjuvanted groups was performed by two-tailed Student’s t-test. Differences were statistically significant (***, 0.001) compared to NP.

Reviewer 4 Report

Manuscript “Protective Efficacy of a mucosal Influenza Vaccine formulation based on the recombinant nucleoprotein co-administered with a TLR2/6 agonist BPPcysMPEG” by Sanchez et al., describes the alternative to current influenza vaccinations. Authors used a TLR2/6 agonist along with the recombinant NP of influenza.  Toll like receptors (TLRs) play a pivotal role in composing immune responses. For eg. the activation of TLR9 signaling exacerbate neurodegeneration by inducing oxidative stress and inflammation, whereas the TLR3 play a key role in several cytokines and chemokines including IFN-β, IFN-γ, TNF-α, IL-1β, and IL-6. Alternative to current influenza vaccination have significant potential as the virus undergoes antigenic shift and drift in each season, affecting the effectiveness of the vaccine.

1.       Authors need to provide more rational for selecting the TLR2/6 agonist in their introduction, even though has been published, it is necessary to the reader to understand the manuscript without going back to other publications.

2.       Authors have used a single influenza virus strain A(H1N1), they should try other strains of virus such as the pandemic one and strain B that are very significant in recent flu seasons. The variability and infection effectiveness among different strains of influenza virus are well known.

3.       In Fig1 and other figures they did not use the BPPcysMPEG alone to show the variation this agonist induce, or do they have any anti-inflammatory effect?

4.       Please define control in all experiments.

5.       Authors may mention the limitations of these compounds in their studies.

6.       The conclusion provided need modification” should be taken into account for the development of innovative vaccines against respiratory viruses” it is difficult to conclude since they have studied only influenza, it should be modified to fit to influenza viruses only.

Author Response

Reviewer 4:

  1. Manuscript “Protective Efficacy of a mucosal Influenza Vaccine formulation based on the recombinant nucleoprotein co-administered with a TLR2/6 agonist BPPcysMPEG” by Sanchez et al., describes the alternative to current influenza vaccinations. Authors used a TLR2/6 agonist along with the recombinant NP of influenza. Toll like receptors (TLRs) play a pivotal role in composing immune responses. For eg. the activation of TLR9 signaling exacerbate neurodegeneration by inducing oxidative stress and inflammation, whereas the TLR3 play a key role in several cytokines and chemokines including IFN-β, IFN-γ, TNF-α, IL-1β, and IL-6. Alternative to current influenza vaccination have significant potential as the virus undergoes antigenic shift and drift in each season, affecting the effectiveness of the vaccine.

  1. Authors need to provide more rational for selecting the TLR2/6 agonist in their introduction, even though has been published, it is necessary to the reader to understand the manuscript without going back to other publications.

We appreciate the reviewer´s comment. As requested by the Reviewer, we have corrected the manuscript to address the reviewer’s criticisms.

“The aim of this study was to develop a universal vaccine of rapid and easy production, capable to induce efficient mucosal and cellular responses which confer protection against influenza infection. For this purpose, the vaccine was designed with the NP antigen, which was obtained as recombinant protein, and it was adjuvanted with a TLR2/6 agonist, the BPPcysMPEG adjuvant [48, 49].

There are some limitations for the parenteral compound Malp-2, such as poor biosolubility in liquid solutions. Moreover, this compound is able to stimulate the immune system by strong induction of various inflammatory mediators, which can influence the biocompatibility. To overcome these limitations, various physicochemical properties and surface modification strategies have been employed to the novel BPPcysMPEG, such as pegylation. In opposite to aluminium-adjuvanted vaccines, which have certain limitations, such as no Th1 reactivity and low stability at low temperaturas, BPPcysMPEG showed high stability at low temperature, enhanced biosolubility in liquid solutions and i sable to induce both antigen-specific cellular (Th1) and humoral (Th2) immune responses.

Moreover, the synthetic TLR2/6 ligand BPPcysMPEG, a pegylated synthetic derivative of the macrophage-activating lipopeptide 2kDa (MALP-2) [50-52], is a powerful adjuvant capable of promoting enhanced immune responses given by mucosal route [53-55]. Previous studies have shown that antigens co-administered with MALP-2 by the mucosal route induced an enhanced B- and T- cell response and improved antigen presentation by dendritic cells, similar or even superior to those observed following parenteral immunization with the same formulations [51, 52, 56, 57]. In comparison to the parent compound MALP-2, BPPcysMPEG showed improved water solubility while retaining its agonistic capacity to stimulate the TLR-2/6 heterodimer [58-61]. Furthermore, MALP-2 has been described to exert beneficial effects on organ damage and the further course after trauma and sepsis [62]. These findings lead us to believe that BPPcysMPEG could be a safe adjuvant candidate for influenza vaccines, which would be capable of inducing mucosal immunity, T-cell responses and humoral immunity. “ Line 106-132

  1. Authors have used a single influenza virus strain A(H1N1), they should try other strains of virus such as the pandemic one and strain B that are very significant in recent flu seasons. The variability and infection effectiveness among different strains of influenza virus are well known.

The reviewer raises a pertinent concern.

We agree that it would have been interesting to perform heterosubtypic challenges.

Unfortunately, we focus first on homologues challenges to evaluate whether the formulation would be a good universal candidate.

Due to changes in animal law in Europe, the studies performed are in accordance with animal ethics based on the international principles of the 3Rs (Replacement, Reduction and Refinement), aiming in the avoidence of unnecessary animal suffering. In this regard, nowadays it is hard to perform additional heterologous challenge studies.

Nevertheless, we rephrased the corresponding paragraph. “Firstly, mice were challenged on day 60 with a sub-lethal dose of the mouse-adapted H1N1 influenza strain A/Puerto Rico/8/34. Mice vaccinated with rNP plus BPPcysMPEG intranasally, showed a better protective efficacy than mice vaccinated with rNP plus BPPcysMPEG by the subcutaneous route, noting that mice vaccinated subcutaneously performed similarly to rNP alone (Suppl. Fig. 3 A/B). It was observed, that mice vaccinated with rNP plus BPPcysMPEG by i.n. route showed no influence on the weight after sub-lethal challenge with homologous influenza strain A/Puerto Rico/8/34 (H1N1). Moreover, animals vaccinated with rNP co-administered with BPPcysMPEG by i.n. route showed reduced morbidity and statistical relevant (***, p< 0.001) gradual weight with only weak weight loss and recovery after 7th days post-infection. Subsequently, a lethal challenge with the homologous influenza strain A/Puerto Rico/8/34 (H1N1) of mice vaccinated with rNP co-administered with BPPcysMPEG was performed to confirm the superior efficacy of the mucosal application route.“ Line 400-412

  1. In Fig1 and other figures they did not use the BPPcysMPEG alone to show the variation this agonist induce, or do they have any anti-inflammatory effect?

We appreciate the reviewer´s comment and as requested by the Reviewer, we have modified the manuscript to address reviewer´s criticisms.

In a preliminary animal study, we did not observe a statistical significant induction of anti-inflammatory IL-10 in supernatants of rNP-restimulated splenocytes when mice were vaccinated with BPPcysMPEG adjuvanted rNP formulation by i.n. route. Moreover, we did not observed any IL-10 secretion in mice vaccinated with BPPcysMPEG adjuvanted rNP formulation by the s.c. route.

Figure: IL-10 secretion in supernatants of antigen-restimulated splenocytes.

  1. Please define control in all experiments.

Control mice were vaccinated with PBS.” Line 166

  1. Authors may mention the limitations of these compounds in their studies.

We appreciate the reviewer´s comment and as requested by the Reviewer, we have modified the manuscript to address reviewer´s criticisms

The aim of this study was to develop a universal vaccine of rapid and easy production, capable to induce efficient mucosal and cellular responses which confer protection against influenza infection. For this purpose, the vaccine was designed with the NP antigen, which was obtained as recombinant protein, and it was adjuvanted with a TLR2/6 agonist, the pegylated Malp-2 adjuvant (BPPcsyMPEG) [48, 49]. Previous studies have shown that antigens co-administered with MALP-2 by the mucosal route induced an enhanced B- and T- cell response and improved antigen presentation by dendritic cells, similar or even superior to those observed following parenteral immunization with the same formulations [51, 52, 56, 57].

There are some limitations for the parenteral compound Malp-2, such as poor biosolubility in liquid solutions. Moreover, this compound is able to stimulate the immune system by strong induction of various inflammatory mediators, which can influence the biocompatibility. To overcome these limitations, various physicochemical properties and surface modification strategies have been employed to the novel BPPcysMPEG. In opposite to aluminium-adjuvanted vaccines, which have certain limitations, such as no Th1 reactivity and low stability at low temperaturas, BPPcysMPEG showed high stability at low temperature, enhanced biosolubility in liquid solutions and i sable to induce both antigen-specific cellular (Th1) and humoral (Th2) immune responses.

Moreover, the synthetic TLR2/6 ligand BPPcysMPEG, a pegylated synthetic derivative of the macrophage-activating lipopeptide 2kDa (MALP-2) [50-52], is a powerful adjuvant capable of promoting enhanced immune responses given by mucosal route [53-55]. In comparison to the parent compound MALP-2, BPPcysMPEG showed improved water solubility while retaining its agonistic capacity to stimulate the TLR-2/6 heterodimer [58-61]. Furthermore, MALP-2 has been described to exert beneficial effects on organ damage and the further course after trauma and sepsis [62]. These findings lead us to believe that BPPcysMPEG could be a safe adjuvant candidate for influenza vaccines, which would be capable of inducing mucosal immuni-ty, T-cell responses and humoral immunity.” Line 106-132

  1. The conclusion provided need modification” should be taken into account for the development of innovative vaccines against respiratory viruses” it is difficult to conclude since they have studied only influenza, it should be modified to fit to influenza viruses only.

We appreciate the reviewer´s comment and as requested by the Reviewer, we have modified the manuscript to address reviewer´s criticisms

The superior immune responses induced by BPPcysMPEG i.n. formulation was consistent with the improved protection efficacy. Mice vaccinated intranasally showed improved protection against infection with the homologous H1N1 strain as compared to those vaccinated with rNP alone. The same is true for mice vaccinated subcutaneously with the adjuvanted formulation.

While our results suggest that the rNP + BPPcysMPEG formulation administered in-tranasally is effective against influenza infection, further experimentation and efforts are needed to make this vaccine a candidate for a universal vaccine. Thus, further studies need to be performed to investigate whether our vaccine candidate is able to stimulate protective immune responses against not only homo-, but also heterosubtypic influenza strains. In this regard, we are aiming in performing immunogenicity and efficacy studies in the highly accepted influenza ferret model.

Regarding the NP antigen used, several clinical trials of NP-containing vaccines have been conducted and, although the use of NP antigen in influenza vaccines is not yet available, it is possible that this antigen will be a strong candidate in universal vaccines in the near future [111, 112]. Another point to bear in mind is that there are few mucosal vaccines on the market, and one of the reasons for this limitation is the lack of effective and safe mucosal adjuvants. Since the COVID-19 pandemics, progress in the use and approval of new vaccines has accelerated dramatically, and there is growing confidence that new mucosal vaccines could be introduced into human vaccines [113]. In this regard, this work clearly demonstrates that BPPcysMPEG is a promising mucosal adjuvant that could be considered for the development of innovative vaccines against respiratory pathogens, such as influenza. “ Line 530-552

Round 2

Reviewer 2 Report

Response to round 2.

I thank the authors for their revised version.

My minor criticisms from my previous review have been addressed.

However, I still have reservations about the response to my main 2 points.

1.       The inadequate description of the antigen.

In the covering letter the authors provide the following information “The nucleoprotein gene derived of influenza strain A/PR/8/34 (H1N1) was cloned into the pET30a plasmid and the protein expressed in Escherichia coli BL21, (DE3) bacteria. The protein was used in previous works of Cargnelutti et al. 1–3. The expression, purification, and characterization of rNP Protein is descripted in detail in previous work 1.”

This is an acceptable level of information. However, I still cannot find anywhere in the manuscript where this is stated.

This must be added.

2.       My main concerns about the inadequacies of the design of the animal studies have not been addressed. The Authors state that local rules governing ethical use of animals have changed to prevent them conducting the required missing comparison of a lethal challenge on vaccine efficacy when administered s.c.

While I have some sympathy with their position, it does not distract from the fact that without this direct comparison it is impossible to draw the conclusion stated on lines 409 – 412 “Subsequently, a lethal challenge with the homologous influenza strain A/Puerto Rico/8/34 (H1N1) of mice vaccinated with rNP co-administered with BPPcysMPEG was performed to confirm the superior efficacy of the mucosal application route.”

The way the data is presented seems designed to minimise this to the reader. If the manuscript is to undergo a further revision, I consider it essential that the data from Supplemental Figure 3 be moved into the main manuscript and incorporated into figure 5.

Throughout the manuscript there is discussion of the different immune responses generated by the vaccine with and without adjuvant when administered s.c or i.n. The interpretation and discussion are often along the lines of i.n. with adjuvant elicits a “better” immune response. This conclusion cannot be draw without the missing animal data.

 I strongly recommend that the authors enter discussion with their respective local ethics committees to point out that the inability to conduct lethal challenge studies is severely hindering their ability to generate meaningful pre-clinical data, both in this study and is likely to do so in future.

As regards the current manuscript. I accept that it is of significance to have shown it is possible to produce protection by this vaccine when administered by the i.n. route in combination with the vaccine. I am willing to change my previous recommendation to reject to one of requires further major revision. But this must include a more balanced interpretation of the data.

Author Response

Reviewer 2: (Revision 2)

We appreciate the comments of reviewer 2 and have corrected the manuscript to address reviewer criticisms.

Specifically:

However, I still have reservations about the response to my main 2 points.

  1. The inadequate description of the antigen.

In the covering letter the authors provide the following information “The nucleoprotein gene derived of influenza strain A/PR/8/34 (H1N1) was cloned into the pET30a plasmid and the protein expressed in Escherichia coli BL21, (DE3) bacteria. The protein was used in previous works of Cargnelutti et al. 1–3. The expression, purification, and characterization of rNP Protein is descripted in detail in previous work 1.” There is no information on the actual antigen beyond it being a recombinant subunit produced in bacteria.

Where is it from?

How is it expressed and purified?

What is the level of purity and how is this determined?

How is structural integrity assessed?

This is an acceptable level of information. However, I still cannot find anywhere in the manuscript where this is stated.

We adapt the text accordingly in the Material & Method part and include additional information.

Vaccine design

“The nucleoprotein gene derived of influenza strain A/PR/8/34 (H1N1) was cloned into the pET30a plasmid and the protein expressed in Escherichia coli BL21, (DE3) bacteria, purified and LPS decontaminated. The recombinant NP was used in previous works of Cargnelutti et al [48, 49].“  Line 139-143

References

  1. Cargnelutti DE, Sanchez M V, Alvarez P, et al. Improved immune response to recombinant influenza nucleoprotein formulated with ISCOMATRIX. J Microbiol Biotechnol. 2012;22(3):416-421. http://www.ncbi.nlm.nih.gov/pubmed/22450799
  2. Cargnelutti DE, Sanchez MV, Alvarez P, Boado L, Mattion N, Scodeller EA. Enhancement of Th1 immune responses to recombinant influenza nucleoprotein by Ribi adjuvant. New Microbiol. 2013;36(2).

  1. My main concerns about the inadequacies of the design of the animal studies have not been addressed. The Authors state that local rules governing ethical use of animals have changed to prevent them conducting the required missing comparison of a lethal challenge on vaccine efficacy when administered s.c.

While I have some sympathy with their position, it does not distract from the fact that without this direct comparison it is impossible to draw the conclusion stated on lines 409 – 412 “Subsequently, a lethal challenge with the homologous influenza strain A/Puerto Rico/8/34 (H1N1) of mice vaccinated with rNP co-administered with BPPcysMPEG was performed to confirm the superior efficacy of the mucosal application route.”

  1. The way the data is presented seems designed to minimise this to the reader. If the manuscript is to undergo a further revision, I consider it essential that the data from Supplemental Figure 3 be moved into the main manuscript and incorporated into figure 5.
  2. Throughout the manuscript there is discussion of the different immune responses generated by the vaccine with and without adjuvant when administered s.c or i.n. The interpretation and discussion are often along the lines of i.n. with adjuvant elicits a “better” immune response. This conclusion cannot be draw without the missing animal data. I strongly recommend that the authors enter discussion with their respective local ethics committees to point out that the inability to conduct lethal challenge studies is severely hindering their ability to generate meaningful pre-clinical data, both in this study and is likely to do so in future. As regards the current manuscript. I accept that it is of significance to have shown it is possible to produce protection by this vaccine when administered by the i.n. route in combination with the vaccine. I am willing to change my previous recommendation to reject to one of requires further major revision.
  3. But this must include a more balanced interpretation of the data.

Reviewer 2 raises a pertinent concern. We agree with the reviewer, that it would have been more appropriate to perform lethal challenge for subcutaneous immunization as well.

  1. We merged Supplentary Figure 3 A/B and Figure 5 A/B as suggested from the reviewer 2.
  2. We modified the text and rephrased the corresponding paragraph in the result part. In a first attempt, mice were challenged on day 60 with a sub-lethal dose of the mouse-adapted H1N1 influenza strain A/Puerto Rico/8/34. Mice vaccinated with BPPcysMPEG by i.n. route showed enhanced protection with almost no influence on the loss weight after sublethal challenge with homologous influenza strain A/Puerto Ri-co/8/34 (H1N1), and statistical significant (***, p< 0.001) compared to rNP alone on day 7 ( 5 C). Notably, the efficacy of the formulation encompassing rNP alone was increased after s.c. application compared to the i.n. vaccination strategy highlighting the necessity of an adjuvant for i.n. vaccination approaches. Mice vaccinated with adjuvanted rNP for-mulation by s,c. route showed a stronger protection level than mice vaccinated with rNP alone, however, differences weren´t statistically significant compared to rNP alone (Fig 5 D). It was observed, that mice vaccinated with rNP plus BPPcysMPEG by i.n. route showed no influence on the weight after sub-lethal challenge with homologous influenza strain A/Puerto Rico/8/34 (H1N1). Moreover, animals vaccinated with rNP co-administered with BPPcysMPEG by i.n. route showed reduced morbidity and statisti-cal relevant (***, p< 0.001) gradual weight with only mild weight loss and recovery after 7th days post-infection.Line 402 – 417

“Subsequently, mice vaccinated with rNP plus BPPcysMPEG intranasally were challenged with a lethal dose of influenza strain A/Puerto Rico/8/34 (H1N1).” Line 417 - 418

Figure 5: XXXXX “Vaccinated BALB/c mice groups were challenged with a sub-lethal dose of 103 ffu/dose of the homo-subtypic influenza strain A/Puerto Rico/8/34 (H1N1) on day 60 by i.n. (Fig 5 C) or s.c. (Fig. 5 D) route. Loss of body weightwas measured daily after challenge for a period of two weeks. SEM are indicated by vertical lines. Statistical analysis between the adjuvanted and non-adjuvanted groups was performed by two-tailed Student’s t-test. Differences were statisti-cally significant (***, 0.001) compared to NP.” Line 424 - 429

  1. We adapt the text accordingly and include additional information in the Discussion part. “The addition of BPPcysMPEG to rNP formulation improved the efficacy of protection compared to vaccination with rNP alone when given by the i.n., as well as by s.c. route, as demonstrated in the sublethal challenge ( 5 C-D). However, the lower efficacy of the rNP alone formulation shown in the i.n. vaccination strategy, highlighted the power of BPPcysMPEG to enhance mucosal protection. This was also verified after the lethal challenge, where all animals were protected. Although the sublethal dose-based infection model showed a clear trend of effective protection due to the s.c. formulation compared to rNP alone, the protective efficacy against lethal infection remains elusive. Thus, even though the infection model based on a sub-lethal dose resulted already in robust efficacy data, it would be important to perform a lethal challenge in a future experiment to measure the protective efficacy of the s.c. formulation.” Line 541 - 551

Reviewer 3 Report

Concerns have been addressed. 

Author Response

We appreciate the comments of all reviewers and modified the text in a 2nd round of revision according to reviewer 2 comments.

Best regards

Reviewer 4 Report

The authors have made significant improvement to the manuscript and the queries are well answered.

Author Response

(The authors gave the same response as above.)

Round 3

Reviewer 2 Report

I thank the authors for taking onboard my comments and making further improvements and clarifications to this manuscript.

I consider that it is now appropriate for publication.

I wish the authors well with their follow up studies on the further development of this vaccine.